# Vitamin D and Glomerulonephritis

**DOI:** 10.3390/medicina57020186

**Published:** 2021-02-22

**Authors:** Guido Gembillo, Rossella Siligato, Michela Amatruda, Giovanni Conti, Domenico Santoro

**Affiliations:** 1Unit of Nephrology and Dialysis, Department of Clinical and Experimental Medicine, University of Messina, 98125 Messina, Italy; rossellasiligato@gmail.com (R.S.); santisi@hotmail.com (D.S.); 2Department of Biomedical, Dental, Morphological and Functional Imaging Sciences, University of Messina, 98125 Messina, Italy; 3Unit of Pediatric Nephrology and Rheumatology, University of Messina, 98125 Messina, Italy; michela.amatruda@gmail.com (M.A.); giovanniconti@hotmail.com (G.C.)

**Keywords:** glomerulonephritis, vitamin D, calcitriol, glomerular disease, VDR, CKD, paricalcitol

## Abstract

Vitamin D presents a plethora of different functions that go beyond its role in skeletal homeostasis. It is an efficient endocrine regulator of the Renin–Angiotensin–Aldosterone System (RAAS) and erythropoiesis, exerts immunomodulatory effects, reduces the cardiovascular events and all-cause mortality. In Chronic Kidney Disease (CKD) patients, Vitamin D function is impaired; the renal hydrolyzation of its inactive form by the action of 1α-hydroxylase declines at the same pace of reduced nephron mass. Moreover, Vitamin D major carrier, the D-binding protein (DBP), is less represented due to Nephrotic Syndrome (NS), proteinuria, and the alteration of the cubilin–megalin–amnionless receptor complex in the renal proximal tubule. In Glomerulonephritis (GN), Vitamin D supplementation demonstrated to significantly reduce proteinuria and to slow kidney disease progression. It also has potent antiproliferative and immunomodulating functions, contributing to the inhibitions of kidney inflammation. Vitamin D preserves the structural integrity of the slit diaphragm guaranteeing protective effects on podocytes. Activated Vitamin D has been demonstrated to potentiate the antiproteinuric effect of RAAS inhibitors in IgA nephropathy and Lupus Nephritis, enforcing its role in the treatment of glomerulonephritis: calcitriol treatment, through Vitamin D receptor (VDR) action, can regulate the heparanase promoter activity and modulate the urokinase receptor (uPAR), guaranteeing podocyte preservation. It also controls the podocyte distribution by modulating mRNA synthesis and protein expression of nephrin and podocin. Maxalcalcitol is another promising alternative: it has about 1/600 affinity to vitamin D binding protein (DBP), compared to Calcitriol, overcoming the risk of hypercalcemia, hyperphosphatemia and calcifications, and it circulates principally in unbound form with easier availability for target tissues. Doxercalciferol, as well as paricalcitol, showed a lower incidence of hypercalcemia and hypercalciuria than Calcitriol. Paricalcitol demonstrated a significant role in suppressing RAAS genes expression: it significantly decreases angiotensinogen, renin, renin receptors, and vascular endothelial growth factor (VEGF) mRNA levels, thus reducing proteinuria and renal damage. The purpose of this article is to establish the Vitamin D role on immunomodulation, inflammatory and autoimmune processes in GN.

## 1. Introduction

Glomerular diseases are the third most common cause of end-stage kidney disease (ESKD) in the United States and represent 25% of chronic kidney disease (CKD) cases in the world [1,2].

In our review, we investigated the connection between Glomerulonephritis (GN) and one of the most used supplements in CKD patients, Vitamin D.

The synthesis of vitamin D active form, the 1α,25-dihydroxyvitamin D3, (calcitriol) takes place mostly in the kidneys by 1α-hydroxylase (CYP27B1), but its action declines as the nephron mass declines [3]. Several mechanisms can stimulate calcitriol renal production: parathormone, low calcium and phosphate serum levels, while elevated phosphate and FGF23 concentrations inhibit its production.

Vitamin D shows pleiotropic effects that encompass skeletal and non-skeletal functions: its active form has the power to modulate the action of renin–angiotensin–aldosterone system (RAAS) [4], stimulate the erythropoiesis [5], can reduce the incidence of cardiovascular events in CKD patients [6], while low vitamin D levels are associated to cardiovascular and all-cause mortality [7].

Patients with renal impairment are at higher risk of Vitamin D deficiency for multiple causes: NS, diabetic nephropathy and GN can cause the loss of its major carrier, the vitamin D-binding protein (DBP); the restriction of nutrients containing Vitamin D to avoid imbalance in phosphorus absorption; the sporadic sunlight exposure [8] and the dysfunction of the cubilin–megalin–amnionless receptor complex in the renal proximal tubule [9].

The Kidney Disease Outcomes Quality Initiative (KDOQI) guidelines recommend that CKD patients who have vitamin D insufficiency (<30 ng/mL) should receive vitamin D supplementation [10].

Vitamin D deficiency is a diffuse issue with high prevalence in CKD, but the best form and posology to prevent and slow the CKD progression and prevent the onset of GN manifestations is still on debate [11].

The primary biomarker of vitamin D status is represented by serum 25(OH)D because it reflects both dietary and ultraviolet radiation influence, presents a longer half-life compared to the active form 1,25(OH)2D and can be measured accurately and reliably. In patients with NS or proteinuric kidney diseases, determination of free 25(OH)D should be preferred to total 25(OH)D levels to diagnose Vitamin D deficiency and establish the therapy [12].

Vitamin D and its receptor (VDR) modulation have been demonstrated to lessen severity of proteinuria in patients with renal impairment, one of GN’s main features [13]. VDR are directly regulated using active Vitamin D such Calcitriol, that also modulates the transcription of several Vitamin D-dependent genes [14].

The best strategies for a correct Vitamin D regulation in CKD and GN are still in discussion, the number of compounds that can be selected for this purpose is rising through the years.

Calcitriol activates VDR directly with an affinity three times higher than that of the Vitamin D analog Paricalcitol, with a ten-times stronger calcemic and phosphatemic power [15]. Doxercalciferol presents similar effects compared with Calcitriol [16], but needs a further hepatic metabolization to be activated. This intermediate step makes Doxercalciferol potentially more modulable compared to the activated form of Vitamin D. This compound is more structurally similar to Vitamin D2, the plant-derived version of Vitamin D, than with the animal-derived version. The Vitamin D3 equivalent of Doxercalciferol is the Alphacalcidol, that is also hydrolysed by the liver in a kidney-independent pathway for its activation.

Vitamin D mimetics such Paricalcitol and Maxacalcitol exerts a milder calcemic effect than Vitamin D active forms. Their bioavailability rises to peak levels and operates on the target tissue following rapid deactivation [17]. This class of drugs decreases the serum levels of intrinsic 1α,25(OH)_2_D, while Vitamin D analogues increases them. Maxacalcitol structurally differs from Calcitriol by the substitution of an oxygen for C22. This modification leads to a reduced affinity of Maxacalcitol for VDR and DBP, guaranteeing a better clearance of this compound [18].

Large comparative studies on different classes of Vitamin D analogs and Vitamin D mimetics conducted in CKD and GN populations are needed to establish the best pharmacological strategies.

Cholecalciferol administration seems to ameliorate albuminuria in CKD patients, even if the data in literature are not conclusive: Molina et al. [19] treated 101 non-dialysis CKD patients with 666 IU/day oral cholecalciferol, with urinary albumin-to-creatinine ratio (uACR) decreasing from 284 (189–425) to 167 mg/g (105–266) at 6 months (geometric mean with 95% CI, *p* < 0.001).

Wu et al. [20] demonstrated that an oral dose of calcitriol (0.25 μg, three times weekly) significantly reduced proteinuria in CKD patients at 8, 16 and 24 weeks of treatment (*p* < 0.05 vs. baseline). A dosage of 0.5 µg calcitriol twice a week has shown efficacy in reducing the proteinuria in patients with IgA nephropathy [21,22] (see Section 4).

In two small randomized controlled trials (RCTs), the investigators demonstrated that patients treated with Paricalcitol had lower urinary protein-to-creatinine ratio (PCR) and 24-h albumin excretion, in comparison to placebo control [23,24].

The VITAL study [25] validated the antiproteinuric effect of the addition of 2 μg/day of paricalcitol to a RAAS inhibitor in diabetic patients: this synergistic effect guaranteed a reduction of residual albuminuria, ranging from –18% to –28% (*p* = 0.014 vs. placebo). In the paricalcitol and ENdothelial fuNction in chronic kidneY disease (PENNY) study [26], 2 μg/d×12 weeks of paricalcitol promoted vasodilatation of vascular smooth muscle and cardiovascular in subjects with CKD stage 3–4. Aperis et al. [27] demonstrated that 1–2 μg daily of Paricalcitol could ameliorate proteinuria in patients with glomerular damage, even if there was a better response in subjects with diabetic nephropathy compared to patients with other types of GN. Other small clinical studies have also shown the potential effects of Paricalcitol in diabetic patients with and without renal involvement [28,29,30].

Vitamin D deficiency is linked to a complex web of severe metabolic abnormalities including inflammation, cardiovascular insults, fibrosis that have far reaching implications for health, leading to progression of renal impairment and ESKD.

The action of the different forms of vitamin D can potentiate the nephroprotective effects of RAAS inhibitors, adding a precious contribution as immunomodulators and anti-inflammatory drugs [31].

## 2. Vitamin D and VDR in Experimental Models of GN

Vitamin D is an efficient endocrine regulator of the RAAS and operates predominantly as a suppressor of renin biosynthesis; on the other hand, dysregulation of VDR leads to elevated renin and angiotensin II production, subsequent hypertension and cardiac hypertrophy [32]. In the kidney, the VDR not only has a major role in the modulation of renin gene expression but is also implicated in the control of inflammation, epithelial-to-mesenchymal transition, and podocyte integrity [33]. Both vitamin D and VDR are involved in the regulation of apoptosis of cultured mouse podocytes and in modulation of transforming growth factor β (TGFβ) via the nuclear factor κB (NF-κB) pathway upon lipopolysaccharide stimulation [34]. In fact, vitamin D demonstrates potent antiproliferative, prodifferentiative, and immunomodulating activities, inhibiting kidney inflammation via VDR-mediated sequestration of NF-κB signaling [35]. Vitamin D inhibits NFκB transactivation also through the modulation of advanced glycation end-products and their receptor (AGE-RAGE system) [36], a mechanism at the basis of progression of different kidney diseases such as diabetic nephropathy, hypertensive nephropathy, obesity-related glomerulopathy, lupus nephritis, amyloidosis, autosomal dominant polycystic kidney disease, and septic acute kidney injury [37].

Vitamin D also contributes to preserving mitochondrial morphology in the renal tissue [38], while VDR activation contributes to mitochondrial integrity, by controlling the permeability transition pore (MPTP) in a ligand-independent way [39]. Mitochondrial preservation is at the basis of cellular function for adenosine triphosphate production, modulation of Ca^2+^ signaling, regulating reactive oxygen species (ROS) status and oxidation-reduction reactions [40].

### 2.1. Calcitriol Use in Experimental GN

Vitamin D has a pivotal role in the kidney’s filtration homeostasis and plays an essential part in podocyte preservation. Podocytes damage and the impairment of the structural integrity of the slit diaphragm have been recognized as a fundamental process in the evolution of glomerulosclerosis [41]. In particular, calcitriol preserves the structural integrity of the slit diaphragm and significantly prevents the loss of nephrin and tight junction protein-1 of rats with membranoproliferative GN [42]. It exerts an antiproliferative effect in course of compensatory growth of nephrons due to subtotal nephrectomy; this action helps to improve glomerular sclerosis and albuminuria [43]. One of its functions is the inhibition of the proliferation of mesangial cells [44] with the capacity of lowering Ki67 mRNA expression and its protein production [45]. Ki67 represents a marker of proliferation, used both as an indicator of excessive cell replication and GN’s progression [46]. Calcitriol found its rationale in experimental studies on IgA nephropathy (IgAN), by the immunomodulation of T helper-regulatory (Th17-Treg) cells balance and by reducing proteinuria in rats [47].

Treatment with Calcitriol can modulate the transient receptor potential cation channel C6 (TRPC6) action in mice’s podocytes, while its deficiency is linked to a dysregulated action of these cation channels, podocyte foot process effacement and proteinuria [48]. These findings have also been demonstrated in diabetic rats: in the study of Zhang et al. [49] treatment with calcitriol increased VDR levels, normalized TRPC6 expression and reduced proteinuria.

Calcitriol also controls the podocyte density through a modulation of mRNA and protein expressions of nephrin and podocin, α3β1 integrin and α/β dystroglycan, contrasting podocyte detachment and podocytopenia [50]. Both podocyturia and nephrinuria are indicators of podocyte damage and markers of worsening of NS; their control has a pivotal role in reducing renal damage progression [51].

Calcitriol also showed a modulatory effect on the urokinase receptor (uPAR), a structure implicated in podocyte damage and development of focal segmental glomerulosclerosis (FSGS). A correct modulation of uPAR has a nephroprotective function, leading to a better control of proteinuria [52,53].

Supplementation with calcitriol has a protective function in controlling the degrading enzyme heparanase expression and subsequent reduction of proteinuria [54]. Garsen et al. [55] demonstrated that through VDR action, calcitriol treatment could regulate the heparanase promoter activity in the podocyte, guaranteeing its preservation.

Yuan et al. [56] revealed in experimental models of IgA nephropathy a positive effect of the association of tacrolimus and vitamin D in the reduction of glomerular mesangial cells hyperplasia, thickening of the glomerular basement membrane, and glomerular infiltration of inflammatory cells, in comparison to placebo and the tacrolimus alone control group. The vitamin D plus tacrolimus group showed a better modulation of NF-κB/TLR4 pathway and reduced levels of TGF-β1, IL-5, and IL-4.

### 2.2. Paricalcitol Use in Experimental GN

Paricalcitol exerts positive effects in the control of proteinuria, glomerulosclerosis, interstitial fibrosis, in the prevention of tubular atrophy and may contrast lymphangiogenesis, another cause of progression of renal disease [57,58]. It also has a cardiorenal protective effect in uremic rats by reducing myocardial fibrosis [59]. Finch et al. [60] confirmed that paricalcitol’s action might be amplified with enalapril addition, decreasing interstitial infiltration of mononuclear cells and oxidative stress, and the association of both drugs is more effective than each compound alone. This pharmacological association has also been tested by Mizobuchi et al. [61] that demonstrated that the use of paricalcitol and enalapril slows the progression of renal insufficiency through the modulation of the TGF-β signaling pathway. Based on this effect, there is the suppression of RAAS gene expression: in fact, paricalcitol lowers angiotensinogen, renin, renin receptor, and vascular endothelial growth factor mRNA status [62]. The association of paricalcitol, enalapril and atrasentan leads to even greater protective power, preventing cardiorenal damage and decreasing cardiomyocyte size to normal levels in uremic rats.

VDR stimulation through the paricalcitol also reduces proteinuria of diabetic nephropathy mice and ameliorate high-glucose-induced injury of kidneys and podocytes [63].

### 2.3. Maxacalcitol in Experimental GN

Maxacalcitol, also known as 22-oxa-calcitriol (OCT), prevents progression of by the control of glomerular volume and glomerular cell number, albumin excretion and rise in creatinine [64]. OCT inhibits mesangial cell proliferation, reducing the mRNA expression of Smooth Muscle Alpha-Actin (alpha-SMA), an actin isoform with a relevant role in fibrogenesis, type I and type IV collagens [65].

The positive effects of OCT on albuminuria and glomerulosclerosis have been also studied in combination with telmisartan [66]. This co-treatment provided a recovery of the slit diaphragm associated proteins with protective effects on podocytes: in fact, it contributes to the restoration of the expression of nephrin, CD2AP and podocin [67].

One of the main differences between OCT and different active forms of Vitamin D, such as calcitriol, is that the first has a lower affinity to DBP, about 1/600 compared to calcitriol [68]. This overcomes calcitriol’s main side effects such as the risk of hypercalcemia, hyperphosphatemia and calcifications; OCT circulates principally as the unbound form with easier availability to target tissues. Hirata et al. [69] showed, in sub-totally nephrectomized rats, that OCT can regulate parathyroid hormone suppression with lower risk of cardiovascular calcification or worsening of residual renal function in comparison with calcitriol.

Sanai et al. [70] demonstrated that an intraperitoneal dose of 0.2 mcg/kg calcitriol three times a week, can accelerate renal deterioration in the course of experimental chronic renal failure, while OCT can attenuate renal histologic lesions. These findings are partially in contrast with the data of Matsui et al. [71] that demonstrated that both treatments with high doses of OCT (2.0 μg/kg/day) and high doses of Calcitriol (0.4 μg/kg/day) have a nephroprotective function, significantly suppressing proteinuria.

### 2.4. Doxercalciferol in Experimental GN

In obese mice, doxercalciferol was effective in decreasing proteinuria, prevented loss of podocyte, decreased mesangial expansion, extracellular matrix protein proliferation, oxidative stress, inflammation and Sterol regulatory element-binding protein 1 and 2 (SREBP-1 and -2), two critical proteins in the control of lipogenesis and mediators of kidney fibrosis [72,73]. Doxercalciferol combined with losartan has a marked power in renin and angiotensinogen suppression: in diabetic mice, it prevents albuminuria, restores glomerular filtration barrier structure and reduces glomerulosclerosis in a dose-dependent manner [74]. The promising use of this compound has been partially assessed in human models, where Doxercalciferol and Paricalcitol demonstrated a lower incidence of hypercalcemia and hypercalciuria than Calcitriol [75].

## 3. Vitamin D in Children with CKD

The importance of appropriate Vitamin D status for kidney outcomes in children has origins during pregnancy: in fact, maternal 25(OH)D levels during pregnancy may influence offspring renal outcomes [76].

Adequate Vitamin D levels in children are associated with a better control of anemia, renal osteodystrophy, secondary hyperparathyroidism, with normalization of Klotho and sclerostin [77,78]. In an analysis of bone metabolism markers in 556 pediatric CKD patients, 25(OH)D levels and serum sclerostin has been demonstrated to be positively associated with a subsequent better control of osteoblast differentiation, proliferation, and activity and reduced osteoblastic bone formation [79].

Most of all, Vitamin D’s preserved function seems to reduce the proteinuria and progression of the renal disease [80]. Despite its pivotal role in renal and systemic homeostasis, low levels of Vitamin D are still a major problem in children with GN; at the same time, Vitamin D intoxication represents a rare but insidious complication of iatrogenic supplementation, leading to hypercalcemia and renal, cardiac, and neurological consequences [81].

Vitamin D deficiency is more common in children and adolescents with late-stage CKD. Vitamin D levels are lower in hypoalbuminemic chilidren and in those with FSGS. In this population, lower 25(OH)D concentrations is also associated with lower circulating 1,25(OH)_2_D concentration [82].

The European Society for Paediatric Nephrology Chronic Kidney Disease Mineral and Bone Disorders and Dialysis Working Groups suggest the use of native Vitamin D for the treatment of Vitamin D deficiency in children with CKD Stages 2–5D with serum 25(OH)D concentrations below 30 ng/mL [83].

A post hoc analysis on 167 children with CKD from the ESCAPE study showed that 25(OH)D levels ≥20 ng/mL are associated with more remarkable kidney function preservation [84]. This value is in line with the Pediatric Endocrine Society and the American Academy of Pediatrics, which established a 25(OH)D threshold of >20 ng/mL as sufficiency in children [85,86].

### 3.1. Vitamin D in Children with Idiopathic Nephrotic Syndrome

The Idiopathic NS is a very common issue in children, with an incidence of is 1.15–16.9 per 100,000 children, depending on ethnicity and country [87].

Nielsen et al. [88] in their study underlined how Vitamin D was insufficient in 93% of pediatric patients at diagnosis of NS. In the study of Pańczyk-Tomaszewska et al. [89], on children with GN, 39% had a deficiency or insufficiency. Zaniew et al. [90] reported a vitamin D deficiency in 16% and vitamin D insufficiency in 75% of children with steroid-treated glomerulopathies, notwithstanding the use of vitamin D supplementation in 82% of the patients.

The treatment with vitamin D and the modulation of its receptor in both adults and children has a central role for immune checkpoint control, being crucial for self-tolerance and the prevention of autoimmunity process based on the most forms of primary and secondary GN. Calcitriol regulates the proliferation of murine and human T cells, producing T cell cytokines, IFN-γ, and IL-17 while inducing IL-4 [91].

1-hydroxyvitamin D3 use in children with both GN and low number of T-lymphocytes in the peripheral blood has been associated with this parameter’s normalization [92]. In fact, T cells are vitamin D targets due to the expression of the VDR, inducing autocrine calcitriol following activation. Moreover, VDR is implicated in developing two cell types, Natural Killer T cells and CD8αα T cells, which inhibit autoimmune uncontrolled processes [93].

Despite the central role of an adequate Vitamin D status in children with GN is well known, the incidence of its deficiency is still high and proper strategies on its correction should be improved.

### 3.2. Vitamin D-Binding Protein in Children with GN

A pivotal role in vitamin D biodisponibility is played by the vitamin D-binding protein (DBP), transporting 85–90% of circulating vitamin D metabolites [94]. In children has been demonstrated that DBP genetic variability affects correlations between vitamin D intake and the clinical measurement of vitamin D status and serum concentrations of its metabolites [95]. Madden et al. [96] showed that critically ill children present low DBP, which is translated not only in a lower biodisponibility of vitamin D and its urinary excretion [97], but also into a lower production of the antimicrobial peptide cathelicidin [98] and higher rate of multiorgan disfunction onset [99].

Urinary Vitamin D-binding protein (uVDBP) has been used as a biomarker of Steroid-Resistant Nephrotic Syndrome (SRNS) in children: its levels were significantly higher (*p* < 0.001) in subjects with SRNS (13,659 ng/mL, interquartile range [IQR] 477–22,979) than in patients with Steroid-Sensitive Nephrotic Syndrome (SSNS) (94 ng/mL, IQR 53–202) [100].

uVDBP has also been tested, in a panel of biomarkers, to identify steroid resistance in the most common glomerular disorder of childhood, the idiopathic nephrotic syndrome (INS). uVDBP independently categorized SRNS vs. SSNS, without relation to proteinuria (AUC, 0.87, *p* < 0.0002) [101]. Choudhary et al. [102] also confirmed the diagnostic utility of uVDBP as a marker of steroid responsiveness in children with ISN: it was significantly higher (*p* < 0.001) in patients with SRNS (701.12 ± 371.64 ng/mL) than in patients with SSNS (252.87 ± 66.34 ng/mL) in comparison to normal controls (34.74 ± 14.10 ng/mL). 

One possible process can be linked to the chronic tubular injury and the dysregulation of intact megalin and cubilin receptors. These molecules guarantee the reabsorption of the filtered DBP in the proximal tubule (PT), with a feedback mechanism, where calcitriol is transported into the human PT epithelium via megalin-mediated endocytosis while bound to DBP [103]. This mechanism is compromised in NS and tubular injury, where the endocytic pathway’s essential components are altered [104].

### 3.3. Vitamin D Receptor Polymorphism in Children with GN

Another essential component of the vitamin D pathway in GN is the VDR activity. VDR has a major role in the prevention of kidney injury by different mechanisms, such as modulation of RAAS activation, hearth diseases, anti-inflammatory role, inhibition of renal fibrogenesis, restoration of mitochondrial function, suppression of autoimmunity and renal cell apoptosis [105,106,107,108]. VDR gene BsmI polymorphism bb genotype is positively linked to NS development [109] and pyelonephritis [110].

Jafar et al. [111] studied four polymorphic sites, *FokI*, *ApaI*, *TaqI* and *BsmI*, among children with INS: they found a significant association for aa of ApaI, BB of BsmI, and ff of FokI, with INS, but no association with the polymorphic sites for TaqI. The combination of the genotypes ff of FokI and BB of BsmI of VDR was associated with a 3.5 increased risk fold. These results are not in line with Al-Eisa et al. [112] that found that both the VDR genetic polymorphisms *TaqI* and *Apal* are not associated with childhood INS. The available data from clinical trials of Vitamin D Receptor Polymorphism in Children with GN are still inconclusive.

## 4. Vitamin D in IgA Nephropathy

IgAN is the most common primary GN worldwide and a significant ESRD cause [113]. Moreover, Vitamin D deficiency is linked to poorer clinical outcomes and more severe IgAN pathological features [114].

The adequate control of proteinuria in IgA patients is essential and represents an indicator of favorable outcome in delaying the disease’s progression [115]. KDIGO guidelines recommend the use of angiotensin-converting enzyme inhibitors (ACEi) or angiotensin receptor blockers (ARB) in IgAN patients with proteinuria above 1.0 g/day [116]. The association of these antihypertensive treatments with Vitamin D for control of proteinuria has been investigated and confirmed in several studies [117,118].

Activated vitamin D has been demonstrated to potentiate the antiproteinuric effect of ARB in IgA nephropathy: patients who received 0.5 μg/day of calcitriol plus Valsartan had a significative proteinuria decrement (from 2.39 ± 0.77 to 1.43 ± 0.57 g/24 h, *p* < 0.01) in comparison to the group treated only with ARB (from 2.46 ± 0.81 to 1.78 ± 0.60 g/24 h, *p* < 0.05) [119].

Szeto et al. [120] showed that Calcitriol may significantly reduce proteinuria and urine PCR decreasing TGF-β level in 10 patients with IgAN with persistent proteinuria notwithstanding the use of ACEi and ARB.

These results have been confirmed by the meta-analysis of randomized trial of Deng et al. [121], which demonstrated that Calcitriol contributes significantly to reduce proteinuria (standard mean difference –1.49, 95% CI (–2.37, –0.62); *p* = 0.0008), with only mild side effects.

In IgAN patients, Calcitriol supplementation showed a stabilization of cardiac autonomic tone in response to an acute physiological stressor, representing a potential preventive cardiovascular therapy in this population [122]. Calcitriol’s cardioprotective role confirms the findings of previous studies that indicate that this compound can modulate the cardiac autonomic system through the enhancement of electrophysiological β-adrenergic signaling between cardiac myocytes [123] or, through VDR control, it can modulate myocardial contractility by the regulation of calcium flux [124].

Moreover, DBPs can be used to evaluate urinary proteome, representing a promising predictive biomarker of responsiveness of IgAN therapy. uVDBPs have 65% sensitivity, 75% accuracy and 85% specificity predicting irbesartan non-responsiveness in IgAN patients [125].

Genetic causes guide the occurrence and progression of IgAN: Shi et al. [126] demonstrated that IgAN progression in subjects with moderate and high genetic risk was 2.12- and 3.61-fold higher than in those with low genetic risk. In support of this concept, Mo et al. [127] studied the relationship between *VDR FokI* single nucleotide polymorphism and renal function in IgA patients: the subjects presenting *VDR FokI* TT genotype had an increased risk of renal dysfunction, demonstrating a potential utility in prediction of the progression of the disease. *VDR* polymorphisms may aggravate IgAN by inhibiting active vitamin D’s protective role, worsening endothelial cells dysfunction, mesangial cell proliferation, and podocyte and tubulointerstitial damage [128]. In this context, VDR influence and its polymorphisms should be adequately and broadly investigated to understand better its predictive role and influence in the progression of IgAN.

## 5. Vitamin D in Lupus Nephritis

Lupus nephritis (LN) is clinically manifest in 50 to 75% of patients with systemic lupus erythematosus (SLE), and 10% of patients with LN will develop ESRD [129]. The majority of SLE patients have suboptimal vitamin D levels; factors that can contribute to this deficit are associated with renal impairment, and the sun protection measures adopted to avoid SLE flares [130]. Vitamin D in patients with LN is significantly lower than in subjects with inactive SLE or active SLE but without LN [131]. These findings are also confirmed in pediatric patients with LN: serum vitamin D levels are inversely correlated with SLE disease activity and LN presence during flares [132]. In vitro experiments showed a nephroprotective role of vitamin D versus LN autoantibodies: serum 25(OH)2D3 was linked with downregulation of aberrant autophagy of podocytes [133].

Pérez-Ferro et al. [134] associated circulating major histocompatibility complex class I-related chain A (MHC-MICA) levels to a subgroup of SLE patients with low vitamin D, innate activation of T cells and LN. This supports the immunomodulatory theory on vitamin D, whom deficit can be linked to the control of cytolytic mechanisms induced by innate immune triggers. Their patients presented an inverse relationship between vitamin D levels and proteinuria (*ρ* = −0.398, *p* = 0.036), and a significant correlation with T-in cell MICA gene expression (*ρ* = 0.526, *p* = 0.008). Vitamin D can help restore the immune system’s homeostasis during SLE flares and contribute to slow LN progression.

VDR stimulation by active vitamin D evokes immune tolerance and promotes the expression of anti-proliferative/pro-apoptotic molecules, and immune cells stimulate the upregulation of VDR transcription [135]. VDR activation stimulates immune cells with intracrine, autocrine, and/or paracrine effects and stimulate different components of both innate and adaptive immunity pathways [136]. In LN, VDR expression promotes regulatory T-cells expansion and reduces the proliferation of Th1 cells, Th17 cells, memory B cells and auto-antibodies, at the basis of SLE [137]. VDR polymorphisms can significantly increase SLE activity and LN related complications [138].

In a study on female Egyptian SLE patients, Emerah et al. [139] found that there is a tight association of VDR *ApaI* AA, *BsmI* BB, and *FokI* FF genotypes with LN and higher SLE activity scores. These findings are in line with the results of Mostowska et al. [140] in Polish SLE patients: subjects with the Ff and FF genotypes of the *FokI* VDR polymorphism presented a major risk of developing kidney disease. Azab et al. [141] also evaluated VDR polymorphism recurrence in the Egyptian population, this time in children and adolescents. In their study there was a significant association between VDR *BsmI* BB genotype with LN (*p* = 0.001). Imam et al. [142] studied *FokI* polymorphism in the same population target, finding a significant association between VDR *FokI* FF genotype with LN (*p* = 0.002).

In an Indian cohort of SLE patients *TaqI, BsmI* and *ApaI* polymorphisms did not significantly associate with LN but interestingly, patients with homozygous polymorphism for *FokI* (ff) and with vitamin D deficiency were exclusively observed among LN cases [143].

In Han Chinese population, VDR gene *BsmI* polymorphism B allele is associated with LN (*p* = 0.027) and also with production of anti-nucleosome antibodies (*p* = 0.037) [144]. These results are confirmed by a study of Ozaki et al. [109] on the Japanese SLE population: they revealed that bb genotype of *BsmI* polymorphism is correlated with NS and renal dysfunction (*p* = 0.0304).

Sun et al. [145] studied VDR tissue expression in renal biopsies of LN patients, founding that it was less represented in renal tubular cells (*p* < 0.001) in LN patients and negatively correlated with activity index (r = −0.548, *p* = 0.012) compared to the control group. DBP has a promising role in nephroprotective strategies in LN patients; low baseline levels of uVDBP are associated with better LN flares outcomes [146].

Go et al. [147] demonstrated that uVDBP is significantly related to proteinuria severity (*p* < 0.001) and renal SLE Disease Activity Index (*p* < 0.001). DBP represented a significant risk factor (hazard ratio 9.627, 95% confidence interval 1.698 to 54.571, *p* = 0.011) for the onset of proteinuric flare in SLE subjects without proteinuria.

## 6. Conclusions

Vitamin D presents a plethora of different functions that goes beyond its role in skeletal homeostasis.

In GN, Vitamin D supplementation demonstrated a significant reduction of proteinuria and beneficial effects in slowing progression of CKD, exerting potent antiproliferative and immunomodulating functions and contributing to inhibition of kidney inflammation. Vitamin D preserves the structural integrity of the slit diaphragm, significantly prevents the loss of nephrin, podocin and tight junction proteins, but also regulates VDR activation, directly reducing renal impairment and guaranteeing many pleiotropic effects [148]. Moreover, activated vitamin D has been demonstrated to potentiate the antiproteinuric effect of RAAS inhibitors in IgAN and LN, enforcing its role in GN treatment. The best strategy to reduce the proteinuria and contrast GN is still to be defined; the results are promising but still not conclusive.

Therefore, we suggest that studies on immunomodulation, inflammatory processes, autoimmune processes, either in vitro on cell cultures or in vivo using lab animals and prospective long-term and larger RCTs, should be conducted to determine all the effects of Vitamin D in these pathologies.

## Data Availability

Data sharing is not applicable to this article as no new data were created or analyzed in this study.

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
