# Peer review of "Vitamin D and Glomerulonephritis"

_medicina, 2021, doi:10.3390/medicina57020186_

Round 1
Reviewer 1 Report
Brief Summary:
The aim of the paper seems to be to review the role of vitamin D supplementation in various formulations and its effect on different glomerulonephritic syndromes. The premise of the paper is to demonstrate the various ways that Vitamin D acts in the body, how deficiency or insufficiency is associated with various GNs and how supplementation could potentially slow progression in renal function decline. The authors review prior research, outlining data that support the premise of their paper and data that contradicts the premise.
Broad comments:
The article is very detailed and descriptive and I enjoyed reading it and learning from it. Overall, good review of prior research regarding benefits of vitamin D with a few side effects mentioned. I would like the authors to clearly state the purpose of this manuscript as it is not mentioned anywhere in the paper. I surmise that the purpose is as stated above in the brief summary but this is an assumption and not clearly stated by the authors. It would be helpful to include a comparison/explanation as to why calcitriol, paricalcitol, maxacalcitol, doxercalciferol have different effects/side effects to one another. I think you did a brief review of this but more information about this would be beneficial and add to the later discussions regarding their use in different GNs.
Specific comments:
Abstract: Would include the purpose of this article/manuscript in the abstract. As it is written, there is no clear objective of the article.
Line 90: “proved” should be changed to “demonstrated” as you cannot definitively prove that paracalcitol could ameliorate proteinuria
Line 92: please clarify wording in regards to “patients with different GN” and avoid using the word “prove” when referring potential effects
Line 117-119: why is this important? Please clarify the significance of this as it relates to GN
Line 141: please clarify “NS.” Do you mean nephrosclerosis?
Line 168: please clarify “DN mice.” Do you mean diabetic nephropathy?
Line 174: what is alpha-SMA?
Line 284: would reword as ACE OR ARB (not “and” since there have been extensive studies showing that combination therapy is not suggested and the KDIGO guideline specifies either therapy not both).
Author Response
Dear reviewer, thank you so much for your comments and suggestions, we updated the latest version of the manuscript thanks to your contribution:
We stated the purpose of the review and added more information on the differences between the Vitamin D compounds. Future researches in the field are still needed but the different characteristics and peculiarities of Vitamin D supplementation and Vitamin D analogs can drive the practitioner to more specific therapies.
We also provided the corrections requested in the specific comments.

Reviewer 2 Report
In the present manuscript, entitled "Vitamin D and Glomerulonephritis", the authors present an overview of the Vitamin D properties in these inflammatory kidney diseases. This is the first review resuming the role of Vitamin D in glomerulonephritis. The topic can represent a relevant contribution to the field. I have only a few minor suggestions for improvement: “Vitamin D” should be capitalized through the text. International standard drug names should not be capitalized (calcitriol for example). Line 157, add the article before “prevention”. Line 283, the spelling of “favourable” is a non-American variant. For consistency, consider replacing it with the American English spelling “favorable”. In the same line, delete “terms of”. Change the “wordy” sentences (for example line 172 use “controlling” instead of “by the control of”. Overall in some sections, there are too many single-sentence paragraphs, which makes it hard to follow. The authors should improve this aspect of the manuscript.” Also, I would expand the introduction section on the functions of Vitamin D Thank youAuthor Response
Dear reviewer, thank you so much for your comments and suggestions, we updated the latest version of the manuscript thanks to your contribution:
We enriched the introduction section with more detailed information and performed the minor corrections as suggested.
Thank you so much for your collaboration

Reviewer 3 Report
Broad comments:
- Overall a comprehensive review of an important issue.
- The discussion about vitamin D in the most common disease in adults (IgA nephropathy) is not paralleled by a discussion about vitamin D in the most common disease in children (idiopathic nephrotic syndrome). The latter is discussed along with CKD in section 3, vitamin D in children with GN. May be the authors can create a separate heading for idiopathic nephrotic syndrome, and on for CKD.
- As for the vitamin D in children with CKD, Kalkwarf HJ et al. in Vitamin D deficiency is common in children and adolescents with CKD, published in Kidney Int 2012; 81:690-697, described vitamin D profile in children with CKD based on their original disease, and Shroff R et al. Clinical practice recommendations for native vitamin D therapy in children with chronic kidney disease stage 2-5 and on dialysis, published in Nephrol Dial Transplant 2017;32:1098-1113 make recommendations for therapy – authors may consider incorporating these two papers in their discussion on page 6, in the paragraph prior to section 3.1.
- Authors could use both units of ng/ml (line 63) and nmol/L (line 217), as the two entries are far apart in the paper, are different (>30 and >50, respectively), and may confuse the reader.
- References section – spaces, fonts and instructions have to be respected – i.e – space between ref 109 and 110, ref 118-119, ref 119-120, font size ref 119; ref 3 and ref 7 – year of publication is bold, to name a few.
- With respect to ref 73, 79 and ref 82 – only the abstracts were in English. Did the authors translate the articles to assess the research described?
Specific comments
- Line 28 – suggest placing “VDR” in parentheses, not “Vitamin D receptor”
- Line 48 – replace “is operated” with “takes place”
- Line 53 – suggest replacing “comprehend” with “encompass” or “include”
- Line 60 – replace “disfunction” with “dysfunction”
- Line 63 – suggest to include <50 nmol/L, as stated above
- Line 64 – what was the authors’ intention with the comment “Vitamin D is a diffuse issue”?
- Line 65 – the statement “and contrast the GN etiopathogenesis” is unclear
- Lines 72-73 – “proteinuria severity” would suggest replacing it with “severity of proteinuria”
- Line 81 – instead of “(see the paragraph vitamin D and IgA nepropathy)”, suggest to refer reader to section 4 – “(see section 4)”
- Line 87 – suggest “paricalcitol” changed to “Paricalcitol”
- Line 91-92 – unclear the part of sentence “…subjects with diabetic nephropathy respect patients with different GN.”
- Line 106 – replace “…still is implicated…” with “…is also implicated…”
- Line 109 – is it necessary for the “nuclear factor κB” to be in bold font?
- Line 110 – after “…stimulation” I suggest to place a “.” and start a new sentence with “In fact…” rather than having one sentence
- Line 171 – instead of “glomerulosclerosis’s progression by the…”, suggest “progression of glomerulosclerosis by its…”
- Line 173 – suggest replacing “expressions” with “expression”
- Line 186 – instead of “microg” suggest either “µg” or “mcg”
- Line 194 – is SREBP-1 and -2 supposed to be bold?
- Lines 206-207 – a better term for “renal osteopathy” may be “renal osteodystrophy”
- Lines 217 and 219 – add >30 ng/ml, as stated in major comments
- Line 269 – VDR gene Bsm1 polymorphism bb was also found in pyelonephritis
- Line 292 - remove extra space between “and” and “urine”
- Line 330 – “This support” should read either “These support” or more likely “This supports”
- Line 374 – “… kidney inflammation inhibition.” could be replaced with “…inhibition of kidney inflammation.”
- Line 387 – duplicate sentence – “All authors have read and agreed to the published version of he manuscript”
Author Response
Dear reviewer, thank you so much for your suggestions, we updated the latest version of the manuscript thanks to your suggestions:
The discussion about vitamin D is now paralleled by a discussion about vitamin D in idiopathic nephrotic syndrome, with a separate paragraph. We also created a paragraph on Vitamin D and CKD in children, enriching it with the articles you suggested.
We used ng/ml instead of nmol/L, uniforming it in the whole text. We also uniformed the references section, respected the spaces, font size and removing the unnecessary bold character. The articles in the references not in English has been discussed with the kind help of native language colleagues.
We also provided the minor modifications requested in the specific comments section.
We attach the updated file below:
